# Genomic Characterization of a Clinical NDM-1-Producing *Klebsiella michiganensis* from Brazil

**DOI:** 10.3390/microorganisms12071408

**Published:** 2024-07-12

**Authors:** Amanda Yaeko Yamada, Andreia Rodrigues de Souza, Amanda Maria de Jesus Bertani, Karoline Rodrigues Campos, Claudio Tavares Sacchi, Denise Brandão de Assis, Enéas de Carvalho, Elizabeth Harummyy Takagi, Marcos Paulo Vieira Cunha, Monique Ribeiro Tiba-Casas, Carlos Henrique Camargo

**Affiliations:** 1Instituto Adolfo Lutz, São Paulo 01246-000, Brazil; amandayy45@gmail.com (A.Y.Y.); andreiarsouza1970@gmail.com (A.R.d.S.); amandabertani94@gmail.com (A.M.d.J.B.); karol_rodriguescamp@yahoo.com.br (K.R.C.); labestrategico@ial.sp.gov.br (C.T.S.); elizabeth.takagi@ial.sp.gov.br (E.H.T.); monique.casas@ial.sp.gov.br (M.R.T.-C.); 2Faculdade de Medicina, Universidade de São Paulo, São Paulo 01246-000, Brazil; cunha.mpv@gmail.com; 3Divisão de Infecção Hospitalar, Centro de Vigilância Epidemiológica, São Paulo 01246-000, Brazil; dbassis@gmail.com; 4Instituto Butantan, São Paulo 05503-900, Brazil; eneas.carvalho@butantan.gov.br

**Keywords:** carbapenem-resistance, conjugation, genetic characterization, IncFIB plasmid, whole genome sequencing

## Abstract

Public health faces daily challenges due to increasing reports of pathogenic microorganisms with new antimicrobial resistance. *Klebsiella michiganensis*, an emerging pathogen, poses difficulty in its identification using conventional techniques. This study presents the first documented case of NDM-1-producing *K. michiganensis* in Brazil, identified as the new ST418. Initially, the isolate from a tracheal secretion was misidentified as *K. oxytoca*. However, accurate identification was achieved through ANI analyses. Whole-genome sequencing was conducted to characterize the genetic context of the resistance genes, to identify virulence factors, and to construct a phylogenetic tree. The *bla*_NDM-1_ gene was found to be harbored on an IncFIB plasmid approximately 112 kb in length, which was transferable in conjugation assays. The detection of carbapenem resistance genes in this species highlights the importance of public health vigilance, as it may serve as a reservoir and disseminator of significant resistance genes.

## 1. Introduction

Antimicrobial resistance presents a significant global threat, constraining treatment options for bacterial infections, thereby diminishing clinical effectiveness, escalating treatment expenses, and increasing mortality rates [1]. It is known that once resistance genes are successfully established in plasmids, they can spread resistance rapidly through different lineages, species, or even genera [2]. *Klebsiella michiganensis*, initially recovered from a toothbrush holder in 2013, is now recognized as an emerging critical human pathogen [3,4]. This species is part of the *K. oxytoca* complex, consisting of nine related species, including *K. oxytoca*, *K. grimontii*, *K. huaxiensis*, *K. michiganensis*, *K. pasteurii*, *K. spallanzanii*, and three unnamed taxa (taxons 1, 2, and 3), which are challenging to differentiate reliably based on their phenotypic characteristics [5]. It has been suggested that *K. michiganensis* may be more clinically relevant in human-associated infections than *K. oxytoca*, but differentiation between these two species is difficult, potentially leading to an underestimated real prevalence of *K. michiganensis* [6,7]. The complex can be divided into phylogroups based on variation in the sequence of the intrinsic gene *bla*_OXY_, encoding a β-lactamase that confers resistance to amino and carboxy-penicillin, of which *K. michiganensis* belongs to the phylogroup Ko1 (with Ko5 representing a sub-lineage) [5]. The presence of a wide range of accessory genes in the pan-genome of *K. michiganensis* suggests an open pan-genome, which indicates its ability to adapt to diverse environmental conditions and its potential genomic plasticity [8].

Moreover, the emergence of carbapenemase-producing *K. michiganensis* poses a new health threat, especially when *bla*-coding genes have the capability to horizontally spread through mobile genetic elements [3]. The resistance in these bacteria primarily involves genes such as *bla*_KPC_, *bla*_NDM_, and *bla*_IMP_, which are associated with plasmids. Strains carrying more than one gene have already been documented in recent years, all co-harboring *bla*_NDM_ with another gene [9,10,11].

Unlike serine-β-lactamases, metallo-β-lactamases require an active site ion and are not inhibited by the so-called second generation of β-lactamase inhibitors, avibactam and vaborbactam, thus impacting antimicrobial therapy [12,13]. Currently, 61 variants of NDM have been identified (according to the BLDL database http://bldb.eu/, accessed on 20 June 2024) [14]. In Brazil, the first *bla*_NDM_ gene was identified in a *Providencia rettgeri* in 2013 and the variant detected was the NDM-1 [15]. According to a recent review, between 2012 and 2021, NDM-producing isolates were identified in 14 bacterial species belonging to eight different genera in Brazil, with 80% belonging to *Klebsiella* spp. and NDM-1 being the predominant allele (54% of isolates) [16]. Another large study identified an increase in the prevalence of NDM-producing organisms in the southern region of Brazil [17].

The *bla*_NDM_ gene is carried by various Inc-group plasmids, such as IncA/C, IncL/M, IncF, IncX, IncHI1A, IncHI1B, and IncN [18,19]. In terms of *K. michiganensis*, there have been few reported cases of strains carrying the *bla*_NDM_ gene, with four from China [9,10,11,20], one from Japan [21], and one from South Africa [22], all of them located in the IncFIB or IncX3 plasmids. To the best of our knowledge, however, there are no reported cases of NDM-producing *K. michiganensis* on the American continents. In the present study, we characterized a clinical *bla*_NDM-1_-producing *K. michiganensis* strain that was isolated in Brazil and explored the phenotypic and genotypic characteristics of the isolate and the plasmid.

## 2. Materials and Methods

### 2.1. Bacterial Identification, Antimicrobial Susceptibility Testing, and PCR to bla_NDM_ Detection

A strain of *Klebsiella* spp. was initially isolated from a tracheal secretion at the local hospital laboratory and sent to our reference laboratory (Bacteriology Center of the Instituto Adolfo Lutz, São Paulo, Brazil) in September 2021. The isolate was identified as ID_1060/21 and submitted for species identification via phenotypic (biochemical series) and molecular tests (MALDI-TOF MS, Bruker Daltonics, Bremen, Germany). The antimicrobial susceptibility test of isolate ID_1060/21 was evaluated by disk diffusion [23] against 21 antimicrobial agents: amikacin (30 µg), amoxicilin-clavulanic acid (20/10 µg), ampicilin-sulbactam (10/10 µg), aztreonam (30 µg), cefepime (30 µg), cefotaxime (30 µg), cefoxitin (30 µg), cefpodoxime (10 µg), ceftazidime (30 µg), ciprofloxacin (5 µg), chloramphenicol (30 µg), ertapenem (10 µg), gentamicin (10 µg), imipenem (10 µg), levofloxacin (5 µg), meropenem (10 µg), norfloxacin (10 µg), piperacillin-tazobactam (100/10 µg), trimethoprim-sulfamethoxazole (1.25/23.75 µg) tobramycin (10 µg), and tigecycline (15 µg). The results were interpreted according to the guidelines described in the Clinical Laboratory and Standards Institute document, M100 [24], and BrCAST 2023, when breakpoints were available. The in-house broth microdilution was performed with cation-adjusted Muller Hinton broth (Sigma-Aldrich, St. Louis, MO, USA) for polymyxin B, in plates ranging from 0.125 to 16 mg/L; MIC values >2 mg/L were considered resistant [25]. A multiplex PCR targeting the *bla*_KPC_, *bla*_NDM_, and *bla*_OXA-48_ genes was performed following established protocols [26]. The strain ID_1060/21 *Klebsiella michiganensis* has been deposited in the Department of Culture Collection of the Adolfo Lutz Institute, a public service collection, as IAL 10144.

### 2.2. Conjugation Assay and Plasmid Size Determination

A conjugation assay with *Escherichia coli* K12 strain J53 (sodium azide-resistant) was performed following Fernandes et al.’s (2017) protocol [27]. The isolates ID_1060/21 and *E. coli* J53 were reactivated on TSA and incubated at 35 °C for 24 h. One colony from each sample was inoculated into LB broth and incubated at 35 °C overnight. The optical density of the two samples was measured and equalized to 0.7 on the McFarland scale, and 100 µL of each was inoculated into the same tube with 3 mL of LB broth and placed at 35 °C overnight. Mueller Hinton plates were prepared with 2.0 µg/mL of imipenem and 100 µg/mL of sodium azide and, on the same day, 100 µL of growth was inoculated onto the plates and spread with a Drigalsk loop, and then incubated at 35 °C for 24 h. Colonies grown on the plates were reseeded onto TSA next to a ceftriaxone disk and isolated colonies grown close to the disk were identified, using MALDI-TOF, as *E. coli*. Of these, a new isolation was performed to obtain pure samples of the conjugate to proceed with the multiplex PCR, to confirm the transference of the resistance gene, and to perform the minimum inhibitory concentration with an E-test with beta lactams.

To determine the size of the plasmid, S1-nuclease pulsed-field gel electrophoresis was performed both in the parental (ID_1060/21) and in the transconjugant strain. The plugs were prepared from a suspension with a transmittance of 4–5%, measured in a spectrophotometer (Analyser, São Paulo, Brazil). Plugs were digested with 0.089 unit of S1-nuclease enzyme (Promega, Madison, WI, USA) for 45 min at 37 °C. For plasmid size determination, a molecular marker, *Salmonella enterica* serotype Braenderup H9812 [28] digested with 30 U XbaI at 37 °C/16 h, was employed. Electrophoresis was performed on the CHEF DR III (BioRad, Hercules, CA, USA) at 6 V/cm and 14 °C, with an initial time switch of 1.0 s and a final pulse of 40.0 s, for 17 h. The gel was stained with ethidium bromide (5 µg/mL) and visualized in a photo documenter (DNR, Bio Imaging Systems, Jerusalem, Israel). Plasmid size was determined using BioNumerics v.8.0 software (Applied Maths, Sint-Martens-Latem, Belgium).

### 2.3. Transconjugant Antimicrobial Susceptibility Testing (AST) 

Minimal inhibitory concentrations (MICs) were determined with an E-test (ampicillin, cefepime, cefotaxime, ceftriaxone, cephalothin, imipenem and meropenem) for the ID_1060/21, the wild *E. coli* J53, and the transconjugant.

### 2.4. DNA Extraction and Whole-Genome Sequencing

Genomic DNA of the strain ID_1060/21 was extracted using the commercial Wizard^®^ Genomic DNA Purification kit (Promega, Madison, WI, USA), starting from an initial volume of 1.4 mL of overnight bacterial growth at 35 °C in LB broth and following the manufacturer’s protocol. Purified DNA was initially assessed using a spectrophotometer from NanoDrop One (Thermo Fisher Scientific, Waltham, MA, USA) and then quantified in a Qubit 4 fluorometer (Thermo Fisher Scientific, Waltham, MA, USA). The DNA’s integrity was verified by E-gel (Invitrogen, Waltham, MA, USA) electrophoresis. 

For short reads’ sequencing using Illumina technology (Illumina, San Diego, CA, USA), the genomic library was prepared using magnetic beads with the Illumina DNA Prep kit to cleave and tag the DNA. To form clusters, a PCR was performed with NexteraTM DNA CD Indexes. Sequencing was carried out at the Strategic Laboratory, Instituto Adolfo Lutz, Brazil, using Illumina MiSeq^®^ equipment to generate 75 bp paired-end reads. For the long reads’ sequencing, libraries were prepared with the Rapid Barcoding Sequencing kit and sequenced using MinION equipment (Oxford Nanopore Technologies, Oxford, UK). 

### 2.5. Genomic Analyzes

FastQC was used to verify the quality of the reads and GC content. The hybrid assembly of the genome’s short and long reads was performed using Unicycler and another assembly was performed, using Flye, for only the long reads [29]. To confirm the identity of ID_1060/21, an Average Nucleotide Identity (ANI) analysis was performed against a local database generated with representative genomes of all *Klebsiella* species available from the NCBI (Appendix A). From the assembled genomes, predicted resistance and gene locations were analyzed using ResFinder, PlasmidFinder, and MGEFinder (all using the Center for Genomic Epidemiology services available at http://www.genomicepidemiology.org/services/ accessed on 1 July 2024), and, to analyze the virulence genes, we employed the Virulence Factor Database [29], which compares the query genome with the references’ accession numbers: AAF37887, NP_752613, NP_286010, and NP_462662. The online tool PathogenWatch (https://pathogen.watch/ accessed on 1 July 2024) was used also for species identification and the determination of sequence length, sequence type (ST), antimicrobial resistance, and the virulence genes’ conference.

For the plasmids comparison, genome sequences were annotated with Rapid Annotation using Subsystem Technology (RAST) and the plasmids’ sequences were compared with BLAST Ring Image Generator (BRIG) software version 0.95. A partial alignment of the genetic environment of the resistance gene was visualized with Clinker (https://cagecat.bioinformatics.nl/tools/clinker accessed on 3 June 2024).

For phylogenetic analysis, 68 genomes (Appendix A) contained in PubMLST (website) were retrieved, together with their metadata, using the CSI phylogeny (http://www.genomicepidemiology.org/services/ accessed on 3 June 2024) tool and visualized in the Microreact platform (http://microreact.org accessed on 3 June 2024).

This Whole-Genome Shotgun project has been deposited in DDBJ/ENA/GenBank under the accession number JAOCNS000000000. The version described in this paper is version JAOCNS020000000.

## 3. Results

The isolate ID_1060/21 was initially misidentified as *K. oxytoca* by biochemical tests and MALDI-TOF MS. Nevertheless, ANI analyses identified ID_1060/21 as *K. michiganensis*, with values ≥ 0.9714 (Appendix A). The presence of chromosomal *bla*_OXY-5_, according to ResFinder, confirms the species’ relationship to the variation in the sequence of this β-lactamase.

*K. michiganensis* ID_1060/21 was resistant to monobactams, penicillins, cephalosporins, tetracycline, and ertapenem, but presented susceptibility to almost all aminoglycosides, fluoroquinolones, phenicol, folate inhibitors, and polymyxin B (MIC 0.5 mg/L) (Table 1). With these phenotypic results, the isolate was classified as having multidrug resistance (MDR) [30].

The multiplex PCR assay was positive only for the *bla*_NDM_ gene. The conjugative transfer of *bla*_NDM_ was successfully achieved to an *E. coli* J53 recipient strain, confirmed by conventional PCR. The transconjugant strain showed more than a 60-fold increase in comparison to the wild *E. coli* J53 (Table 2). S1-PFGE revealed that isolate ID_1060/21 carries three plasmids sized 163.49 kb, 105.91 kb, and 50.52 kb, but only the 103.55 kb plasmid was transferred according to the S1-PFGE analysis (Appendix A).

Hybrid assembly successfully identified the presence of 10 contigs in a chromosome length of 6,071,905 bp. Three additional contigs were assembled, corresponding to the plasmids observed in S1-PFGE.

*K. michiganensis* ID_1060/21 was identified as the new sequence type ST418 (allelic profile: *gapA* 3; *infB* 50; *mdh* 15; *pgi* 22; *phoE* 18; *rpoB* 6; *tonB* 4). Sequencing analyses showed the presence of *bla*_NDM-1_, found in an IncFIB(K) plasmid, named pID_1060/21, ~112,325 bp in size, in accordance with the observed plasmids detected in the S1-PFGE of both the parental and transconjugant strain (Appendix A). The beta-lactam resistance genes *bla*_OXY-5_ (chromosomal gene) and *bla*_TEM-1_ (IncN2 plasmid) were also detected. ID_1060/21 presented four virulence genes with >80% identity (accession numbers NP_752613, AAF37887, NP_286010, and NP_462662, respectively): *entB* (siderophores enterobactin), *ompA* (outer membrane protein), *yagZ/ecpA* (common pilus structural subunit), and *mgtB* (magnesium-transporting).

The sequence of the pID_1060/21 plasmid was blasted against the NCBI database, and eight different plasmid sequences (Appendix A) were necessary to almost cover the entire 112 kb sequence (Figure 1). Nevertheless, some regions were not identified in other plasmids (as highlighted in yellow in the center circle in Figure 1).

The region of pID_1060/21 containing the *bla*_NDM-1_ gene presented similarity with the plasmid p_kv_NDM1 (from *K. variicola* BM_DM_115, isolated from a blood sample in Bangladesh, 2016; accession CP095680.1), an IncFII(K)/IncFIB(pQil)-type plasmid (Figure 2). The IS*Aba125*-like (IS*30* family) truncated and IS*Sup2*-like (IS*630* family) element were identified as the mobile genetic elements located upstream of the gene *bla*_NDM-1_. Downstream, the *bla*_NDM-1_ gene, *ble*_MBL_, *trpF*, *dsbD*, and IS*15DII*-like (IS*6* family) element were identified (Figure 2).

Based on the analysis of the *K. michiganensis* pubMLST collection, 43 different STs were identified among the 68 evaluated isolates retrieved from 14 countries: Australia (n = 13), Brazil (n = 2), Canada (n = 3), Denmark (n = 5), England (n = 6), France (n = 2), Ireland (n = 2), Israel (n = 7), Italy (n = 15), Norway (n = 7), Russia (n = 3), Spain (n = 1), the USA (n = 1), and unknown (n = 1) (Appendix A). This shows that there is a diversity to clones all over the world, with no ST predominating. An association of clones to certain regions was observed, such as ST354 in Denmark, ST376 in Norway, ST254 in Israel, and ST95 in Italy (Figure 3).

## 4. Discussion

In this study, we report the first case in Brazil of *K. michiganensis* harboring the *bla*_NDM-1_ gene within a transferable IncFIB plasmid. The emergence of carbapenemase-producing *K. michiganensis* highlights its potential role as a reservoir and disseminator of clinically relevant resistance genes [9]. Noteworthy cases of *K. michiganensis* carrying NDM variants have been reported globally in recent years, spanning regions such as China [9,10,11,20], South Africa [22], and Japan [21].

Similar to our isolate, Zhang et al. (2022) identified an NDM-producing *K. michiganensis* in China presenting *bla*_NDM-1_ located in a 233,442 bp IncFIB(K)/IncFII(K) plasmid (Genbank accession: JAHNZR000000000.1) [9]. Conversely, Zheng et al. (2018) recovered a similar isolate from a stool sample carrying *bla*_NDM-1_ in a 106,140 bp IncFIB/IncFIIY plasmid (Genbank accession: NZ_CP022350.1) [11]. Notably, both studies detected multiple carbapenemase genes, including *bla*_KPC-2_, *bla*_NDM-1_, and *bla*_IMP-4_ [9] and *bla*_KPC-2_, *bla*_NDM-1_, and *bla*_NDM-5_ [11] in a single *K. michiganensis* strain, underscoring its potential as a reservoir for disseminating different resistance genes to other pathogens.

Diverging from our findings, previous reports of carbapenemase NDM in *K. michiganensis* have identified different Inc-type plasmids. NDM-1 was located in IncX3 plasmids in South Africa [22] and China [10], while NDM-5 was associated with IncX3 plasmids in Japan [21] and China [11]. The presence of resistance genes such as *bla*_NDM_ and its variants in diverse plasmids may elucidate the broad dissemination of these enzymes [17], thus posing a significant public health challenge.

An in-depth analysis of the genetic context of the *bla*_NDM-1_ gene revealed a conserved triad of genes (*ble*_MBL_ *trpF*, and *dsbD*) downstream of *bla*_NDM-1_, corroborating the existing literature [31].

According to the phylogenetic tree, ID_1061/21, identified as the new ST418, was closely associated with two isolates from Russia, both classified as ST349. Pan-genomic analyses conducted by Simoni et al. (2023) revealed that 35% of *K. michiganensis* genes are part of the core genome, emphasizing the existence of a diverse array of accessory genes within this species [8]. According to the authors, this suggests potential genomic plasticity linked to adaptation in novel environments, as well as the continuous microevolution of *K. michiganensis*. An escalating number of reports document *K. michiganensis* co-harboring genes implicated in carbapenem resistance. In a Chinese study encompassing 25 carbapenem-resistant *K. oxytoca* complex strains, *K. michiganensis* emerged as the predominant species in 16/25 (64% of isolates), with 5 harboring *bla*_NDM_ resistance genes and an additional 10 strains containing *bla*_KPC_ or *bla*_IMP_ [32].

Since the initial identification of NDM in Brazil, numerous publications have spotlighted carbapenemase occurrences in *Enterobacterales*. This study represents the first report of NDM-producing *K. michiganensis* in the country, elucidating the plasmid and genetic context of the *bla*_NDM-1_ gene harbored by a transferable IncFIB plasmid. The increasing incidence of *K. michiganensis* carrying, accumulating, and disseminating resistance genes globally underscores the critical need for reinforced vigilance, aiming to identify and control the dissemination of carbapenemase-producing pathogens in healthcare settings.

## Figures and Tables

**Figure 1 microorganisms-12-01408-f001:**
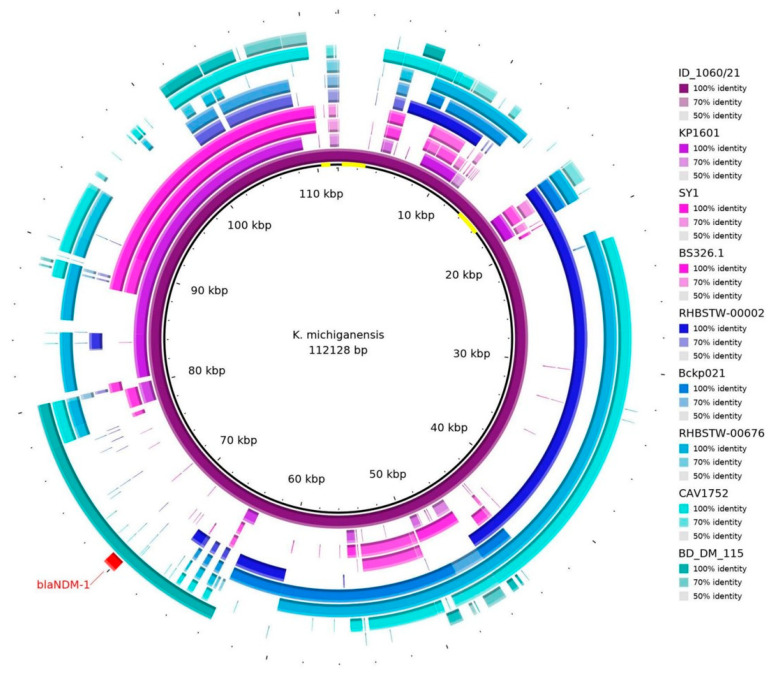
Plasmid hits, according to blastn (https://blast.ncbi.nlm.nih.gov/Blast.cgi), against eight sequences in the database from the NCBI. It is possible to see that the *bla*_NDM-1_ gene (in red) is aligned with the BD_DM_115 sequence (plasmid p_kv_NDM1; accession: CP095680.1). Some regions were out of alignment with other sequences, as shown in yellow in the center of the representative circle.

**Figure 2 microorganisms-12-01408-f002:**
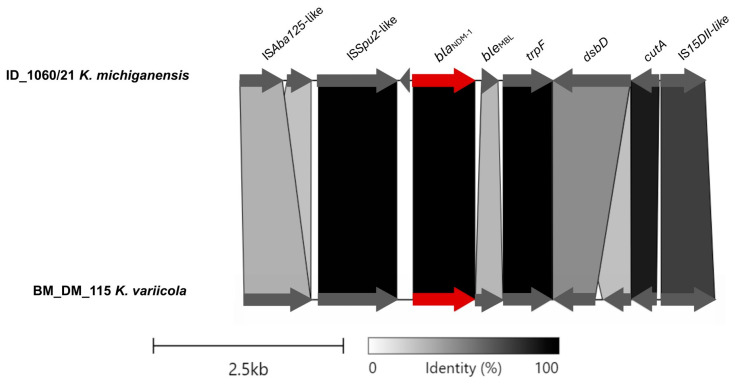
Partial alignment of ID_1060/21 and BD_DM_115 plasmids and the linearized comparison of the *bla*_NDM-1_ genetic environment, with *bla*_NDM-1_ highlighted in red. This figure was created using the Clinker tool (available at https://cagecat.bioinformatics.nl/tools/clinker accessed on 3 June 2024).

**Figure 3 microorganisms-12-01408-f003:**
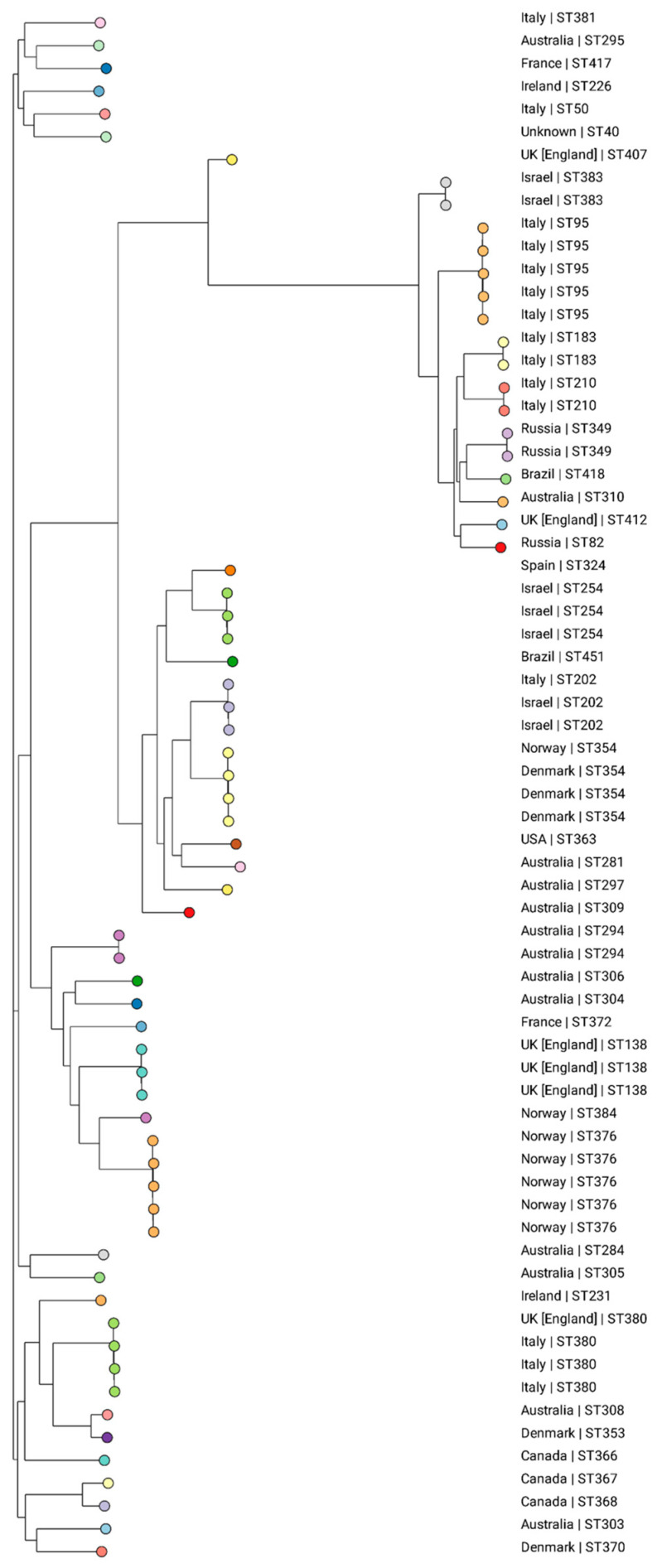
Phylogenetic tree generated from the analysis of the SNPs of 68 global *K. michiganensis* isolates using CSIPhylogeny, visualized on the Microreact website. The colors represent the different STs and the country of isolation is presented for each strain. Isolate ID_1060/21 is clustered with two isolates from Russia (ST349).

**Table 1 microorganisms-12-01408-t001:** Antimicrobial susceptibility testing, determined by disk diffusion, for ID_1060/21 and categorizations according to CLSI 2023.

Classes of Antibiotics	Antibiotics (Concentration)	Categorization According to CLSI	Categorization According to BrCAST
Monobactams	Aztreonam (30 µg)	Resistant	Resistant
Penicillins	Amoxicillin-clavulanic acid (20/10 µg)	Resistant	Resistant
	Ampicillin-sulbactam (10/10 µg)	Resistant	Resistant
	Piperacillin-tazobactam (100/10 µg)	Resistant	NA
Cephalosporins	Cefepime (30 µg)	Resistant	Resistant
	Cefotaxime (30 µg)	Resistant	NA
	Ceftazidime (30 µg)	Resistant	NA
	Cefpodoxime (10 µg)	Resistant	Resistant
	Cefoxitin (30 µg)	Resistant	Resistant
Carbapenems	Ertapenem (10 µg)	Resistant	Resistant
	Imipenem (10 µg)	Intermediate *	Susceptible
	Meropenem (10 µg)	Intermediate *	Intermediate *
Aminoglycosides	Gentamicin (10 µg)	Resistant	Resistant
	Tobramycin (10 µg)	Susceptible	Susceptible
	Amikacin (30 µg)	Susceptible	Susceptible
Fluoroquinolones	Ciprofloxacin (5 µg)	Susceptible	Susceptible
	Levofloxacin (5 µg)	Susceptible	Susceptible
	Norfloxacin (10 µg)	Susceptible	Susceptible
Phenicol	Chloramphenicol (30 µg)	Susceptible	NA
Folate inhibitor	Trimethoprim-sulfamethoxazole (1.25/23.75 µg)	Susceptible	Susceptible
Glicilcycline **	Tigecycline (15 µg)	NA	Resistant

* Susceptible to increased exposure; ** Tigecycline: breakpoint according to BrCAST 2023. NA: not available.

**Table 2 microorganisms-12-01408-t002:** Results of comparative MICs between *E. coli* J53 receptor, transconjugant, and ID_1060/21.

Isolate	AMP	FEP	CTX	CRO	CFL	IMP	MEM
*E. coli* J53	2	0.32	0.015	0.015	3	0.125	0.016
Transconjugant	>256	>16	>256	>32	>256	12.0	1.0
ID_1060/21	>256	>16	>256	>32	>256	8.0	3.0

AMP: Ampicillin; FEP: Cefepime; CTX: Cefotaxime; CRO: Ceftriaxone; CFL: Cephalothin; IMP: Imipenem; MEM: Meropenem.

## Data Availability

This Whole-Genome Shotgun project has been deposited in DDBJ/ENA/GenBank under the accession number JAOCNS000000000. The version described in this paper is version JAOCNS020000000. The strain ID_1060/21 was deposited in the Department of Culture Collection of the Adolfo Lutz Institute and is available as IAL 10144.

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
