# Peer review of "Genomic Characterization of a Clinical NDM-1-Producing Klebsiella michiganensis from Brazil"

_microorganisms, 2024, doi:10.3390/microorganisms12071408_

Round 1

Reviewer 1 Report

Comments and Suggestions for Authors

The authors report the first case of NDM-1-producing Klebsiella michiganensis in Brazil. This study addresses a significant global issue concerning antibiotic resistance. Overall, the manuscript is well-written and well-organized. The authors comprehensively characterize this strain, effectively covering the topic.

 Please maintain consistent bacterial nomenclature: Use the full name first, followed by the abbreviation (revise it in full text).

  Lines 49-50: I suggest rewriting this sentence to replace 'plasmid genes,' as these genes can also be found on the chromosome

My suggestion: “The resistance in these bacteria primarily involves genes such as blaKPC, blaNDM, and blaIMP, which are associated with plasmids”.

Author Response

Manuscript ID: microorganisms-3092256

Title: Genomic characterization of a clinical NDM-1-producing Klebsiella

michiganensis from Brazil

Thank you for the opportunity of having our manuscript revised.

We have read all the suggestions, and we made the corrections in the manuscript. All the changes are highlighted in red.

Reviewer 1

The authors report the first case of NDM-1-producing Klebsiella michiganensis in Brazil. This study addresses a significant global issue concerning antibiotic resistance. Overall, the manuscript is well-written and well-organized. The authors comprehensively characterize this strain, effectively covering the topic.

Thank you for your time and attention revising our manuscript.

 Please maintain consistent bacterial nomenclature: Use the full name first, followed by the abbreviation (revise it in full text).

We revised the text to address your suggestion.  

  Lines 49-50: I suggest rewriting this sentence to replace 'plasmid genes,' as these genes can also be found on the chromosome

My suggestion: “The resistance in these bacteria primarily involves genes such as blaKPC, blaNDM, and blaIMP, which are associated with plasmids”.

Thank you for the suggestion. The manuscript was changed to address your comment.

Reviewer 2 Report

Comments and Suggestions for Authors

The submitted manuscript presents the detection and genetic context of the blaNDM-1 gene identified for the first time in K. michiganensis in Brazil, an emerging pathogen. The article is succinct, presents the results adequately and discusses them well. Below are some considerations:

Lines 80 - 83: It suggests adding antibiotic concentrations in each disk.

Lines 83 - 85: What was the reason for using CLSI as a guide instead of BrCAST, since this study was conducted in Brazil?

Lines 87: Add PB concentrations range.

Line 89: Until then, the isolate had been identified as K. oxytoca by MALDI-TOF. Is this information correct? The authors were able to identify it more precisely after analyzing the genome. Please make this information clear in the methodology, although it is explicit in the Abstract.

Line 149: What is the reference strain used to investigate virulence factors in VFDB?

Figure 1 and 2: Improve image quality.

Figure 2: Highlight the blaNDM-1 gene.

Author Response

Reviewer 2

The submitted manuscript presents the detection and genetic context of the blaNDM-1 gene identified for the first time in K. michiganensis in Brazil, an emerging pathogen. The article is succinct, presents the results adequately and discusses them well. Below are some considerations:

Thank you for your time and attention revising our manuscript.

Lines 80 - 83: It suggests adding antibiotic concentrations in each disk.

Concentration of each disk was included in the section 2.1.

Lines 83 - 85: What was the reason for using CLSI as a guide instead of BrCAST, since this study was conducted in Brazil?

We initially used the CLSI breakpoints to make our data comparable with other studies, but we agree with the reviewer and we included in this version the interpretation of breakpoints by BrCast, considering that this information is important for Brazilian audience (Table 1).

Lines 87: Add PB concentrations range.

PB concentration range was included in this version.

 Line 89: Until then, the isolate had been identified as K. oxytoca by MALDI-TOF. Is this information correct? The authors were able to identify it more precisely after analyzing the genome. Please make this information clear in the methodology, although it is explicit in the Abstract.

Thank you for this observation. We included this information in the first line of results.

Line 149: What is the reference strain used to investigate virulence factors in VFDB?

According to the VFDB database, the sequences of reference strains presenting the accession Numbers AAF37887, NP_752613, NP_286010 and NP_462662 were employed. This information was included in our revised manuscript.

Figure 1 and 2: Improve image quality.

Figures were made to achieve the 300dpi quality. Please see the attached files instead of the figures in the body text.

Figure 2: Highlight the blaNDM-1 gene.

Figure was revides, as well as its legend.